# How Did Life Emerge in Chemically Complex Messy Environments?

**DOI:** 10.3390/life12091319

**Published:** 2022-08-26

**Authors:** Kenji Ikehara

**Affiliations:** 1G&L Kyosei Institute, The Keihanna Academy of Science and Culture (KASC), Keihanna Interaction Plaza, Lab. Wing 3F, 1-7 Hikaridai, Seika-cho, Souraku, Kyoto 619-0237, Japan; ikehara@cc.nara-wu.ac.jp; Tel.: +81-774-73-4478; 2International Institute for Advanced Studies, Kizugawadai 9-3, Kizugawa, Kyoto 619-0225, Japan

**Keywords:** GADV hypothesis, origin of life, protein 0th-order structure, origin of protein, [GADV]-microsphere, origin of gene: the core life system

## Abstract

One of the problems that make it difficult to solve the mystery of the origin of life is determining how life emerged in chemically complex messy environments on primitive Earth. In this article, the “chemically complex messy environments” that are focused on are a mixed state of various organic compounds produced via prebiotic means and accumulated on primitive earth. The five factors described below are thought to have contributed to opening the way for the emergence of life: (1) A characteristic inherent in [GADV]-amino acids, which are easily produced via prebiotic means. [GADV] stands for four amino acids, Gly [G], Ala [A], Asp [D] and Val [V], which are indicated by a one-letter symbol. (2) The protein 0th-order structure or a [GADV]-amino acid composition generating water-soluble globular protein with some flexibility, which can be produced even by the random joining of [GADV]-amino acids. (3) The formation of versatile [GADV]-microspheres, which can grow, divide and proliferate even without a genetic system, was the emergence of proto-life. (4) The [GADV]-microspheres with a higher proliferation ability than others were able to be selected. Proto-Darwin evolution made it possible to proceed forward to the creation of a core life system composed of the (GNC)_n_ gene, anticodon stem-loop tRNA or AntiC-SL tRNA (GNC genetic code), and [GADV]-protein. (5) Eventually, the first genuine life with a core life system emerged. Thus, the formation processes of [GADV]-protein and the (GNC)_n_ gene in chemically complex messy environments were the steps to the emergence of genuine life.

## 1. Introduction

Human beings have tried for many years to determine how life emerged on primitive Earth. In other words, human beings are interested in the origin of life. In addition, there is another significance to life-origin research. For example, answers to fundamental about genes and proteins, such as “why are there four types of nucleotides or bases in RNA and DNA?”, and similarly, “why are modern proteins made up of 20 types of amino acids?”, could be obtained if the mystery of the origin of life was solved. However, irrespective of the strenuous efforts of many researchers, the mystery of the origin of life remains unsolved. The main reasons for this are as follows.

**A.** Difficulties elucidating the establishment process of the fundamental life system

The most important point for solving the mystery of the origin of life is to clarify the fundamental life system establishment process, which involves six members (gene, genetic code, tRNA, metabolism, cell structure and protein) [1] (Chapter 2). However, it has been difficult to understand this establishment process for the following reasons.

1. The “chicken–egg relationship” between genes and proteins: The so-called “chicken–egg relationship” has made it difficult to solve the mystery of the origin of life for many years. In such a situation, the RNA world hypothesis, which assumes that first life arose from the RNA world, which was formed by the self-replication of RNA, was proposed by Gilbert in 1986 [2]. However, it would be principally impossible to solve the mystery from the standpoint of this hypothesis because any gene encoding a mature protein could never be formed in the absence of a target protein or in the absence of protein, even if RNA could be produced on primitive Earth.

On the other hand, I proposed the [GADV]-protein world hypothesis (in short, the GADV hypothesis) approximately 20 years ago [1,3,4], assuming that the first life emerged from the [GADV]-protein world, which was formed by the pseudo-replication of [GADV]-proteins [5]. I believe now that the mystery could be solved based on the GADV hypothesis. [GADV] represents four amino acids, Gly [G], Ala [A], Asp [D] and Val [V]. The square brackets ([ ]) are used to discriminate the one-letter symbols of amino acids, particularly Gly [G] and Ala [A], from the one-letter symbols of nucleobases guanine G and adenine A.

2. The emergence of life under random processes: This theory suggests that any occurrences on primitive Earth proceeded as random processes. However, it is easily supposed that a mature gene and a mature protein with an ordered sequence could never be formed through a random process at one stroke, because the sequence diversities of gene encoding a protein composed of 100 amino acids and the protein itself are extraordinarily large at (4^3^)^100^ = ~10^180^ and 20^100^ = ~10^130^, respectively [1] (Chapter 3) [6]. Regarding this problem, I believe that the question of how a gene acquired the genetic information of a mature protein could be solved if the question were considered from the standpoint of the GADV hypothesis [1] (Chapter 8), as explained in detail later.

**B.** The problem of a research strategy based on experiments

1. The mystery of the origin of life could not be solved with experiments only: Needless to say, experiments are quite important or even crucial in studying the origin of life. However, it would be impossible to solve the mystery with experiments only, because the events that occurred on primitive Earth approximately 4 billion years ago could never be reproduced by experiments carried out in a present-day laboratory.

2. The establishment process of the core life system, composed of gene, genetic code (tRNA) and protein [1] (Chapter 2), could not be determined using bottom-up approaches only: In order to clarify the origin of life, it is undoubtedly important to answer the questions regarding what happened on primitive Earth to lead to the emergence of life. Therefore, many researchers have conducted studies for many years according to bottom-up approaches in a bid to discover where and what kind of organic compounds were produced on primitive Earth. However, it would probably be impossible to solve the mystery of the origin of life this way, because the establishment process of the core life system would never be made clear by simply using bottom-up approaches, unless newly born life or RNA/DNA containing the most primitive genetic information for protein synthesis could be found from rocks approximately 4 billion years ago.

3. The difference between an experimental condition in a laboratory and a primitive Earth situation: Experiments confirming nucleotide synthesis on primitive Earth have been carried out by many researchers, and results showing that nucleosides were actually produced with ribose and nucleobases are frequently reported in scientific journals [7,8,9,10,11,12]. However, it seems that it would be impossible to apply these results to the real occurrences on primitive Earth, as the experimental conditions, such as the concentrations and purity of reactants, are always very different from the situation on primitive Earth. Therefore, I consider that a parallel use of both bottom-up approaches, which have been carried out thus far, and top-down approaches, for example, using database analyses of modern genes and proteins, could hold one of the keys to solving the mystery of the origin of life [13].

**C.** The issue of how genes and proteins, which are composed of relatively small types of components, could be produced in chemically complex messy environments on primitive Earth

As described in the “Special Issue Information” of the Special Issue (Origin of Life in Chemically Complex Messy Environments), “Considering the prebiotic Earth four billion years ago (a messy atmosphere, in other words), a chaotic mélange of diverse starting materials appears realistic”, there is another problem that makes it difficult to solve the mystery of the origin of life. Biopolymers as genes and proteins, composed of only four types of nucleotides and twenty types of amino acids, respectively, are used in extant organisms. However, such biopolymers must have been formed in chemically complex messy environments on primitive Earth. Therefore, how such biopolymers, using small types of components, could be formed in the messy environments of primitive Earth is an issue. When the mechanism producing genes and proteins, which uses a relatively small types of components, was acquired by something, that something would become the first life. Therefore, understanding the steps to the emergence of life should lead to the discovery of the correct answer to the question of how genes and proteins could be produced in chemically complex messy environments.

In this paper, I would like to discuss the processes of how biopolymers, composed of rather small types of relatively simple monomers, were formed in chemically messy environments from the standpoint of the GADV hypothesis [1,3,4]. For this purpose, it is necessary to understand the processes through which first life emerged on primitive Earth. Then, I will explain only the main points regarding how the GADV hypothesis could be proposed, as my ideas on the origin of life have already been described in detail in the book, *Towards revealing the origin of life: Presenting the GADV hypothesis*, which was published last year by Springer [1]. Thereafter, I will explain how genes and proteins were formed during repeated random reactions in chemically messy environments using small types of the respective components, and I will attempt to answer the third question (C) regarding how genes and proteins could have been produced using small types of, respectively, selected monomers from the messy environments.

Therefore, this article is described as providing an answer to the question that has been proposed in the Special Issue: “Origin of Life in Chemically Complex Messy Environments”. In other words, I discuss in the article how a small types of amino acids or nucleotides were selected from the chemically complex messy environments of primitive Earth based on the GADV hypothesis, which I propose. The answer to the question described in the article is the novelty aspect of this paper.

## 2. The Key to Solving the Mystery of the Origin of Life (Protein 0th-Order Structure: [GADV]-Amino Acids)

The most important concept in the GADV hypothesis is one of protein 0th-order structures or [GADV]-amino acid composition [14]. First, I will explain the significance of protein 0th-order structure in solving the mystery of the origin of life (Figure 1).

Consider here the problem based on the GADV hypothesis, how a gene, which encodes one amino acid sequence of a mature protein, was created. It is well known that it is impossible to produce one mature protein like a precise polymer machine through the random joining of even simple [GADV]-amino acids at one stroke, because the amino acid sequence diversity of a protein composed of only four types of one hundred [GADV]-amino acids is extraordinarily large (4^100^ = ~10^60^) [1] (Chapter 3).

How were the genes that encoded mature proteins formed? For this purpose, one immature water-soluble globular protein with some flexibility, produced by the expression of one double-stranded (ds)-RNA encoding one essentially random [GADV]-amino acid sequence in the protein 0th-order structure, is indispensable for the formation of the gene (Figure 2). An RNA with a random (GNC)_n_ codon sequence could be formed by the random joining of GNC anticodons carried by AntiC-SL RNAs [1] (Chapters 7 and 8).

Specifically, every gene that encodes a mature protein has been formed as a result of maturation from an immature or incomplete water-soluble globular protein with some flexibility, which generates various weak catalytic activities or demonstrates pluripotency [1] (Chapter 3), to a mature protein with a rigid structure and high catalytic activity.

This means that the formation of a mature protein always requires an immature protein, because the formation of a mature protein must always be led by the elevation of the weak catalytic activity of the immature protein, which is easily understood as the relationship between a key and a key hole (Figure 2). This is one of the reasons why the mystery of the origin of life has not been solved using the RNA world hypothesis; that is, RNA never acquires genetic information for mature protein synthesis independently of an immature protein, even if RNA could first be produced by a random process or by the random joining of nucleotides in prebiotic environments.

As described above, both an immature [GADV]-protein and ds-RNA are required to form the first RNA gene (Figure 2). At present, an essentially random codon sequence similar to a random (SNS)_n_ codon sequence or a non-stop frame on an antisense strand of a GC-rich gene is used as the field for creating an entirely new gene [15]. On primitive Earth, immature [GADV]-proteins were produced using a [GADV]-amino acid sequence encoded by a random (GNC)_n_ codon sequence on one of two RNA strands, which were formed by the random joining of GNC anticodons carried by four AntiC-SL tRNAs (Figure 2) [1] (Chapters 3 and 8) [4]. The reason it was possible is because one amino acid sequence randomly selected from a [GADV]-amino acid pool (protein 0th-order structure) is essentially the same as one amino acid sequence arranged by a random (GNC)_n_ RNA sequence. Thus, the GADV hypothesis, which I have proposed, is an idea based on one of the protein 0th-order structures or [GADV]-amino acid composition.

## 3. Possible Steps from Chemical Evolution to the Emergence of First Life

Next, explicitly consider the steps to the emergence of first life on primitive Earth according to the GADV hypothesis (Figure 3) [1]. The reason why the steps to the emergence of life must be described here is that these steps are intimately related to the establishment process of the core life system composed of genes, genetic code (tRNA) and proteins, and to the formation process of proteins composed of four types of [GADV]-amino acids, which were encoded by the (GNC)_n_ RNA gene composed of four types of nucleotides. In other words, the emergence of a genuine life was intimately related to the formation process of the (GNC)_n_ RNA gene and [GADV]-protein using a relatively small type of components in chemically messy environments on primitive Earth.

Next, I explain the main object of this article regarding how biopolymers (genes and proteins) composed of small types of the respective monomers were formed in chemically messy environments on primitive Earth.

In this section, the steps from chemical evolution to the emergence of the first genuine life, equipped with the fundamental life system composed of six members, are discussed as divided into three (Table 1): the first part—from chemical evolution to the formation of the [GADV]-microsphere (proto-life); the second part—from the formation of the [GADV]-microsphere to the formation of AntiC-SL RNA; and the third part—from AntiC-SL RNA to the emergence of the first genuine life. The processes are discussed in three parts because the aspect of selection or of evolution changed before and after the formations of the [GADV]-microsphere. In addition, the second and third parts are separated because the use of nucleotides is restricted for the first time into four, A, G, U and C, by the formation of AntiC-SL RNA, which is folded with base pairs, A-U and G-C. On the other hand, the number of amino acids could not still be restricted to four at the time point as the establishment of the core system, composed of genes, tRNA (genetic code) and proteins, made it possible to use the four [GADV]-amino acids for the first time.

In addition, only random processes should have been naturally repeated before the first genuine life emerged or in the absence of any gene encoding an ordered amino acid sequence of a [GADV]-protein. Therefore, it is also essential to clarify how [GADV]-proteins with an ordered amino acid sequence were formed or how genes encoding the amino acid sequence were formed during the repetition of random processes in order to solve the mystery of the origin of life.

### 3.1. Part I. Steps from Chemical Evolution to Formation of [GADV]-Microsphere ([GADV]-Protein World)

Many studies on chemical evolution have been carried out to clarify what types of organic compounds can be synthesized from inorganic compounds under what conditions [16,17]. In these studies, the types of biomolecules that can be synthesized, such as amino acids, sugars, nucleobases, etc., which are necessary for first life to emerge, were mainly investigated.

As can be easily understood from previous studies, messy organic compounds, including various organic acids and amines in addition to various non-natural amino acids, should be produced via prebiotic means. Steps to the emergence of life proceeded through reaction processes, with the compounds selected from messy organic compounds. However, it would be quite difficult to select out only the necessary organic compounds, for example, only [GADV]-amino acids, from messy mixtures of organic compounds (Figure 4). In this Part I, I aim to explain a principle that can advance towards a solution to the problem from perspective of the GADV hypothesis.

#### 3.1.1. Preferential Synthesis of [GADV]-Amino Acids via Prebiotic Means

In previous studies, it was confirmed that simple organic compounds, such as amino acids and nucleobases with a small number of carbon atoms, were synthesized via various prebiotic means, as described below:(1)Electric discharge into the primitive atmosphere (Miller-type experiments) [16,17];(2)Catalytic reactions on pyrite, clay, hydrothermal vents, and so on [18,19,20];(3)Organic synthesis via heavy bombardments of meteorites or asteroids [21,22];(4)Introduction of organic compounds via meteorites, asteroids, space dust, and so on, from space [23,24].

Thus, various organic compounds including, especially, oxygen atom(s) were synthesized through physical and physico-chemical reactions, and messy compounds were introduced from space, accumulating on primitive Earth.

Amino acids were produced with organic compounds, which were selectively synthesized and accumulated in large amounts on primitive Earth via prebiotic means. Note that, at this point in time, a type of selection among organic compounds had been carried out during the synthetic processes of the organic compounds based on the nature of the chemical compounds themselves used as reactants, as described below.

##### Preferential Synthesis of α-Amino Acids

Amino acids were produced using prebiotic methods more easily than fatty acids and hydrocarbons, of which constituent atoms are connected chiefly with inactive carbon-carbon and carbon–hydrogen bonds. In fact, Miller describes in his book that there is no good method for fatty acid synthesis via prebiotic means [16]. On the contrary, glyoxylate and pyruvate must be rather easily produced via prebiotic means, as those keto-acids carrying a small number of carbon atoms have active carbon atoms with a localized electron, although it must be confirmed experimentally that glyoxylate and pyruvate could be synthesized via prebiotic means. Furthermore, it is well known that α-amino acids were obtained in larger amounts than β-amino acids in Miller’s experiments [16]. These results indicate that α-amino acids were rather easily synthesized via prebiotic means and accumulated in large amounts.

#### 3.1.2. Direct Random Joining of [GADV]-Amino Acids

Described next is the mechanism or principle of how only [GADV]-amino acids could eventually be selected out at a relatively high rate for primeval protein synthesis in the messy mixture of various organic compounds containing non-natural amino acids. The reasons why α-amino acids could be preferentially selected for the synthesis of protein are as follows.

[GADV]-proteins, actually aggregates of [GADV]-peptides, could be also produced via prebiotic means in messy organic compounds, which accumulated in large amounts on primitive Earth, although the proteins were incomplete in the sense that various amino acids other than [GADV]-amino acids were contained in the proteins.

##### Preferential Polymerization of [GADV]-Amino Acids

Amino acids should be selectively linked with each other during repeated wet-drying processes, as amino acids have both positive and negative charges in the molecules to facilitate easy association in water and to create a peptide bond between two amino acids [25,26]. Nevertheless, various organic compounds other than [GADV]-amino acids were naturally incorporated into [GADV]-proteins or [GADV]-peptide aggregates during the direct random joining of [GADV]-amino acids.

##### Preferential Association of [GADV]-Peptides Containing Val

All four [GADV]-amino acids with a rather simple structure were easily synthesized via prebiotic means. However, the synthetic amount of Val, which has a more complex molecular structure, should be much less than that of the other three amino acids, Gly, Ala and Asp. Nevertheless, the lesser amount of Val could be compensated for through the formation of [GADV]-peptide aggregates, owing to the high hydrophobicity of Val, because peptides containing a larger amount of Val could be preferentially associated with each other through large hydrophobic interactions. This also contributed to the formation of more active immature [GADV]-proteins.

#### 3.1.3. Preferential Synthesis of [GADV]-Amino Acids by Immature [GADV]-Proteins

It should be considered that various organic compounds, especially [GADV]-amino acids, could be preferentially synthesized using simple organic compounds with functional groups as glyoxylate and pyruvate, even by immature [GADV]-proteins with weak catalytic activities, and [GADV]-proteins also could be produced by the random joining of [GADV]-amino acids under the protein 0th-order structure [1] (Chapters 3, 5) [14]. On the contrary, it must be difficult to synthesize hydrocarbons with immature [GADV]-proteins. It is considered that the selective synthesis of [GADV]-amino acids by immature [GADV]-proteins, which was performed before cell structure formation, further advanced the steps to the emergence of life at a faster rate than before.

On the other hand, many inactive and useless [GADV]-peptides could also be produced during the direct random joining of [GADV]-amino acids, because of the incorporation of various organic compounds into the peptides. In the reaction, [GADV]-peptides with sufficiently high catalytic activity could always be produced as a result of a wide distribution of [GADV]-peptides, which were synthesized through a random process, although the formation rate of active [GADV]-proteins might be low [26]. This made it possible to proceed towards the emergence of life at a faster rate than in the era of chemical evolution without immature [GADV]-peptide catalysts.

#### 3.1.4. Incorporation of Non-Natural Amino Acids into Immature [GADV]-Proteins

Various α-amino acids and β-amino acids, other than [GADV]-amino acids, such as 2-aminobutylic acid (2-ABA), 2-aminopentanoic acid, β-alanine and so on, could be also produced via prebiotic means. Therefore, various amino acids other than [GADV]-amino acids would also be incorporated into immature [GADV]-proteins during the polymerization of [GADV]-amino acids, because non-natural amino acids could not be effectively excluded during simple polymerization among amino acids with both positive and negative charges in the molecule. Thus, it would be significant to form [GADV]-microspheres facilitating chemical evolution, as described below.

### 3.2. Part II. Steps from Formation of [GADV]-Microsphere to AntiC-SL RNA Formation

#### 3.2.1. Significance of Cell Structure as a [GADV]-Microsphere for Facilitating Chemical Evolution

##### Incorporation of Various Organic Compounds into [GADV]-Microspheres

After sufficient amounts of [GADV]-amino acids accumulated on primitive Earth, [GADV]-microspheres were formed, for example, by repeated wet-drying processes in depressions of rocks on primitive Earth [1] (Chapter 4). The [GADV]-microspheres inevitably contained large amounts of [GADV]-peptides in the cell structure so that the [GADV]-peptides were synthesized by immature [GADV]-proteins in the microspheres. The supposition that the immature [GADV]-proteins could synthesize [GADV]-peptides is supported by the fact that even Gly-Gly and Gly-Gly-Gly have peptide synthetic activity [27]. The formation of [GADV]-microspheres made it possible to hold oligomeric [GADV]-peptides, which were synthesized in the microsphere, owing to the semi-permeable [GADV]-protein membrane. The accumulation of [GADV]-peptides in the [GADV]-microsphere generated higher osmotic pressure to induce the further incorporation of low molecular weight organic compounds as glyoxylate and pyruvate into the microsphere (Figure 5).

However, such proteins, into which amino acids other than [GADV]-amino acids were incorporated at a higher rate, would be gradually excluded, because [GADV]-microspheres using [GADV]-peptides composed of a higher rate of [GADV]-amino acids would be selected at a higher probability during proliferation followed by evolution. The selection became possible because [GADV]-amino acid composition is one of the protein 0th-order structures, and the incorporation of non-natural amino acids into [GADV]-proteins caused malfunction of the [GADV]-proteins. For the same reason, 2-ABA was excluded from natural amino acids as [GADV]-amino acids in order to avoid the duplicate use of Ala and 2-ABA, both of which are α-helix-forming amino acids [28]. The most important aspect of [GADV]-microsphere formation would be growth and proliferation, induced by high osmotic pressure accompanied by [GADV]-peptide synthesis (as shown in Figure 6).

##### Selection of [GADV]-Microspheres with a High Proliferation Ability

In fact, at first, incomplete [GADV]-peptides containing non-natural amino acids would be produced at a high probability. However, [GADV]-microspheres containing non-natural amino acids at a lower rate could grow, proliferate and evolve faster than others to leave more descendants even before the establishment of the genetic system, because the [GADV]-microspheres containing at a higher rate of [GADV]-amino acids could acquire [GADV]-proteins with the higher function necessary to proceed to the emergence of life (Figure 6). Thus, [GADV]-microspheres containing lesser amounts of non-natural amino acids than others were consequently selected and proliferated. Furthermore, the formation of [GADV]-microspheres or [GADV]-protein membranes made it possible to protect against the dissipation of [GADV]-amino acids into environments, because immature [GADV]-proteins, produced with [GADV]-amino acids, could not ooze out through the [GADV]-protein membrane. This also contributed to a more efficient chemical evolution.

##### Growth, Division, Proliferation and Inactivation of [GADV]-Microspheres

[GADV]-microspheres with a higher ability for growth, division and proliferation could be consequently selected out from among many [GADV]-microspheres, as described above (Figure 6). The selected [GADV]-microspheres could leave more descendants and evolve further, even if the microspheres did not hold any genetic system. Contrasted with that, many other microspheres, which were defeated in the struggle for existence, disappeared, leaving many inactive bodies, for example, in the depressions of rocks on primitive Earth. However, those inactive bodies were reused for the growth and prosperity of the selected [GADV]-microspheres. This situation is similar to that observed on the present Earth where components of withered plants and dead bodies of animals are usually reused by presently living organisms after they have been degraded by various organisms, including bacteria.

##### Formation of Proto-Metabolic Pathways for [GADV]-Amino Acid Synthesis

As described above, [GADV]-amino acids with a sufficiently high stability were easily produced and accumulated in large amounts on primitive Earth. In addition, it was easy to refill [GADV]-amino acids, even if the amino acids were exhausted upon consumption for growth, because [GADV]-amino acids could be synthesized with simple organic compounds as glyoxylate and pyruvate, which accumulated on primitive Earth in large amounts and could be easily supplied from the environments into the microspheres (Figure 5) [1] (Chapter 5). Thus, [GADV]-amino acids were optimal compounds for advancing the steps to the emergence of life on primitive Earth.

Even such [GADV]-amino acids, which supported the proliferation of [GADV]-microspheres, would be depleted from inside of the microspheres and the proliferation of the [GADV]-microspheres would terminate soon after the deprivation due to the exponential growth of [GADV]-microspheres. The only way to avoid the situation was the construction of a proto-metabolic system for the synthesis of [GADV]-amino acids in the microspheres that were growing exponentially [1] (Chapter 5).

Proto-metabolic reactions using immature [GADV]-proteins started, in [GADV]-microspheres, to produce [GADV]-amino acids and [GADV]-peptides, just after the formation of [GADV]-microspheres. This would have been supported by the fact that [GADV]-amino acids can be synthesized using glyoxylate and pyruvate as the starting materials in a few reaction steps in modern metabolic pathways (Figure 7A). Inversely, the cycle of growth, division and proliferation of [GADV]-microspheres would terminate if the proto-metabolic pathways were not formed in the [GADV]-microspheres, and if sufficient osmotic pressure could not be maintained. Therefore, it can be considered that only [GADV]-microspheres, which had a high synthetic ability to produce [GADV]-peptides, were evolutionally selected and could leave more descendants than others. From these considerations, it can be concluded that the cell structures, which were indispensable to selection and evolution and in which [GADV]-peptides were synthesized to maintain high osmotic pressure, is the most essential function for life, not the genetic system.

As is well known, modern cell membranes are composed of phospholipids and membrane proteins. Membrane proteins, but not phospholipids, exhibit various membrane functions. On the contrary, phospholipids are used for filling the gaps among membrane proteins and for expediting the migration of membrane proteins to express membrane functions more efficiently. Therefore, it would be valid to consider that phospholipids were inserted into [GADV]-protein membranes after the formation of phospholipid synthetic pathways via proto-metabolism in [GADV]-microspheres.

##### Formation of Proto-Metabolic Pathways for Nucleotide Synthesis

On the contrary, nucleotides, which are necessary to produce RNA, could not be synthesized in large amounts via prebiotic means because of the complex chemical structures of nucleotides. It can be seen from previous studies that nucleotides could not be produced by Miller’s experiments [16,17] and that nucleotides have not been found in meteorites. It is still controversial whether or not nucleotides and nucleosides were actually synthesized using the reactants of simple inorganic compounds via prebiotic means, although some experimental results show that nucleotides and nucleosides could be produced with ribose-5-phspate or ribose and nucleobases, respectively, have been reported [9,29,30], because the concentrations and purity of the reactants used in the experiments are usually quite different from the conditions on primitive Earth.

Even papers showing that nucleosides and nucleotides were synthesized from formamide with meteorite catalysts under proton irradiation, were published [31,32]. Nevertheless, nucleotides must be depleted from the surroundings of [GADV]-microspheres, which proliferated exponentially. This means that proto-metabolism for nucleotide synthesis must be established in [GADV]-microspheres before deprivation of nucleotides. To establish the core life system, sufficient amounts of nucleotides necessary to synthesize RNA must be produced with previously existing immature [GADV]-proteins. For this purpose, proto-metabolic pathways, through which nucleotides could be synthesized, must be formed (Figure 7B). Needless to say, the formation of metabolic pathways in the absence of genes must rely on random processes.

Therefore, I consider that nucleotides could be produced through proto-metabolism using glyceraldehyde as a starting compound for ribose 5-phosphate synthesis [1] (Chapter 5). It is supposed that glyceraldehyde, having three electronically localized carbon atoms, could accumulate in large amounts, similar to glyoxylate and pyruvate, although it must be confirmed that glyceraldehyde could be synthesized via prebiotic means. Furthermore, nucleotide synthetic pathways could be formed with immature [GADV]-proteins because, in addition to the pluripotency of the immature [GADV]-proteins [1] (Chapter 3), [GADV]-microspheres, which acquired more favorable metabolic pathways for proliferation than others, even accidentally, could leave more descendants than others. I would like to name this phenomenon as “proto-Darwin evolution”. In this way, during the evolutionary process, metabolic pathways, including nucleotide synthesis, which were favorable for the proliferation of [GADV]-microspheres, were formed in the microspheres.

However, many researchers may consider that nucleotides could never be synthesized from glyceraldehyde with immature [GADV]-proteins if they do not know the significance of the pluripotency of immature [GADV]-proteins, which could be synthesized by the direct random joining of [GADV]-amino acids under the protein 0th-order structure [1] (Chapter 3). Therefore, I would like to stress that the mystery of the origin of life will never be solved as long as they rely on nucleotide synthesis via prebiotic means only. Similarly, it would be reasonable to consider that the RNA world could never be formed on primitive Earth, as it would be impossible to produce a sufficient amount of RNA leading to the emergence of life without nucleotide metabolic pathways.

#### 3.2.2. Formation of [GADV]-aa-AntiC-SL tRNA

##### Evolution of Activated [GADV]-Amino Acids

Initially, the synthesis of [GADV]-peptides in [GADV]-microspheres was carried out with [GADV]-amino acids by immature [GADV]-proteins, which were produced by the direct joining of [GADV]-amino acids, such as through wet-drying processes in the depressions of rocks on primitive Earth [25,26]. However, [GADV]-peptides could be produced at a much faster rate by using activated [GADV]-amino acids as [GADV]-adenosine monophosphate ([GADV]-AMP) than with the direct use of [GADV]-amino acids. Subsequently, activated [GADV]-amino acids were used for the more efficient synthesis of [GADV]-peptides in the order below, although there was no difference between the direct joining of [GADV]-amino acids and [GADV]-peptide synthesis with activated [GADV]-amino acids, except the difference in reaction rate.

1-1. Use of [GADV]-aminoacyl (aa)-AMP: [GADV]-aa-AMPs were synthesized with immature [GADV]-proteins (actually [GADV]-peptide aggregates) to accelerate peptide synthesis after adenosine triphosphate (ATP) was synthesized and accumulated in large amounts in [GADV]-microspheres. At this point in time, it is supposed that the activated [GADV]-amino acids were exclusively used for peptide synthesis with immature [GADV]-proteins owing to the accumulation of ATP in the microspheres

1-2. Use of [GADV]-aa-3′-ACC: [GADV]-aa-3′-ACCs were successively used for the synthesis of [GADV]-peptides. The stability of single-stranded (ss)-trinucleotide, CCA-3′, against the RNase activity of immature [GADV]-proteins made it possible for use in the synthesis. It is easy to understand that the ss-3′-ACC is stable against the RNase activity, because the 3′-ACC end of modern tRNA is also stable against RNase. Needless to say, the synthesis of [GADV]-peptides with activated [GADV]-amino acids was carried out non-specifically, because such activators, as AMP and 3′-ACC, cannot generate specificity to the respective [GADV]-amino acids. However, the use of [GADV]-aa-3′-ACCs contributed to a more efficient synthesis of [GADV]-peptides, as more sites of 3′-ACC than ATP, itself, could be used for binding with immature [GADV]-protein enzymes.

Of course, not only amino acids, but also other organic compounds might be activated with ATP and 3′-ACC. In the case of peptide synthesis with activated amino acids too, messy organic compounds could be incorporated into the peptides (Figure 4). However, it is supposed that activated organic compounds other than [GADV]-amino acids did not eventually contribute to the emergence of life, because the functions of [GADV]-peptides containing meaningless organic compounds would be lowered.

1-3. Use of 3′-ACC-AntiC-SL RNA: After the use of 3′-ACC, [GADV]-peptide synthesis was carried out using 3′-ACC-AntiC-SL RNA (Figure 8). One of the reasons why AntiC-SL RNA was used for the activation of [GADV]-amino acids is that AntiC-SL RNA, which was produced during cycles of oligonucleotide synthesis and degradation of the oligonucleotides, was the smallest but was a sufficiently stable RNA against hydrolysis by immature [GADV]-proteins [33]. In addition, the association of two AntiC-SL tRNAs side by side through base pairing between U and A in the two AntiC-loop RNAs facilitated the peptide bond formation between two amino acids bound to the 3′-ACC-end [1] (Chapter 7).

The AntiC-SL RNA primitive tRNA hypothesis is supported by the fact that base pairs between two complementary GNCs are stable [34], as well as the fact that any base in an anticodon loop of three AntiC-SLs, except Asp-tRNA modified with queuosine and 2-methyladenosine, out of four modern *Escherichia coli* [GADV]-aa-tRNAs is not chemically modified [35].

### 3.3. Part III. Steps from Formation of AntiC-SL RNA to the Emergence of Life

#### 3.3.1. Formation of ds-(GNC)_n_ RNA Gene

##### Establishment of the Core Life System in [GADV]-Microspheres

Of course, the initial [GADV]-proteins, which were produced with immature [GADV]-proteins in a [GADV]-microsphere, were not literally [GADV]-proteins, meaning that [GADV]-proteins contained non-natural amino acids other than [GADV]-amino acids. The synthesis of such impure polypeptides should always occur before the formation of the first gene or in the absence of genes on primitive Earth. However, even the impure and immature [GADV]-proteins could advance catalytic reactions in the microspheres, although the activities were low. Therefore, literal [GADV]-proteins could not be produced until the GNC primeval genetic code and the (GNC)_n_ gene were formed. Inversely, the selection of [GADV]-microspheres with purer [GADV]-protein with higher catalytic activity made it possible to form the first gene, leading to the synthesis of pure [GADV]-proteins. Consequently, the genetic system or the core life system composed of the (GNC)_n_ gene, AntiC-SL tRNA (GNC code) and [GADV]-protein was invented to improve [GADV]-protein functions through the complete exclusion of non-natural amino acids [1] (Chapter 6). Thus, the genetic system was formed to establish the most primitive, but pure, [GADV]-protein synthesizing system.

Needless to say, the first gene encoding the first mature [GADV]-protein with an ordered amino acid sequence must have been generated through an evolutionary process containing at least one random process, as described below.

(1)Synthesis of ss-(GNC)_n_ RNA: In this case, the key point is to understand the process of how the first ss-(GNC)_n_ RNA was formed, because an RNA with a random (GNC)_n_ codon sequence could be formed by the random joining of GNC anticodons carried by the four AntiC-SL RNAs, although it must be confirmed that ss-(GNC)_n_ RNA could be formed as I expected. [1] (Chapters 7 and 8). Note that the synthetic process of an immature [GADV]-protein with a random [GADV]-amino acid sequence through a random (GNC)_n_ codon sequence is essentially the same as [GADV]-protein synthesis by the direct joining of [GADV]-amino acids. Then, how was the ss-(GNC)_n_ RNA formed? I consider the formation process as follows [1] (Chapter 8).

1-1. Two pairs of two AntiC-SL tRNAs, which were bound in a column using two complementary GNC anticodons, were further aligned side-by-side to make a tetramer of four AntiC-SL RNAs (Figure 8).

1-2. Two anticodons of the two AntiC-SL tRNAs, which were aligned side-by-side, were connected by a phosphodiester bond. Thus, a random (GNC)_n_ RNA sequence encoding a random [GADV]-amino acid sequence was created (Figure 8).

However, many researchers may doubt whether the ss-(GNC)_n_ RNA could be formed as described above. My answer to the question is as follows.

1-3. The consecutive codons on mRNA have been actually read by two anticodons of two adjacent modern tRNAs.

1-4. The serial reading mechanism of the codon sequence on mRNA with the anticodon of tRNA clearly indicates that it is reasonable from a stereochemical viewpoint to consider that a ss-codon sequence (mRNA) can be produced by the joining of tRNA anticodons. That is the reason why a successive codon sequence can be read by the anticodons of two tRNAs tightly bound side-by-side. However, it would be difficult for two tRNAs to read a successive codon sequence if the codon sequence was formed independently of tRNA. The genetic sequence on mRNA was translated by AntiC-SL tRNA as the reverse process of the formation of the (GNC)_n_ codon sequence in [GADV]-microspheres (Figure 8) [1] (Chapter 8). Thus, the fact that a comma-less codon sequence can be translated by tRNAs indicates that a genetic sequence was formed by the random joining of the anticodons carried by tRNAs.

Note that the random (GNC)_n_ codon sequence is the simplest but most meaningful sequence, able to be produced through a random process and also able to be used for immature [GADV]-protein synthesis. This also indicates that the formation of a (GNC)_n_ codon sequence via the random joining of the GNC anticodons of AntiC-SL tRNA is only one way under which an immature but meaningful [GADV]-protein could be produced through the RNA sequence, formed by an essentially random process.

(2)Formation of a ds-(GNC)_n_ RNA: Successively, a ds-(GNC)_n_ RNA was formed by the complementary strand synthesis of the ss-(GNC)_n_ RNA.(3)Formation of the first ds-(GNC)_n_ RNA gene: Finally, the first (GNC)_n_ RNA gene was formed. The formation of the first (GNC)_n_ gene was triggered by the synthesis of an immature [GADV]-protein from one strand of the ds-RNA. The (GNC)_n_ codon sequence encoding a random [GADV]-amino acid sequence evolved to one ds-RNA gene encoding a mature [GADV]-protein as led by the promotion of the activity on the immature [GADV]-protein (Figure 2 and Figure 8).

Thus, ds-(GNC)_n_ RNA genes encoding a mature [GADV]-protein were formed as the most effective means for producing mature [GADV]-proteins with high catalytic activity. It should be noted again that the steps towards the formation of the first gene were the processes for selecting only [GADV]-amino acids more efficiently from chemically complex messy environments. Thus, the key to the first ds-(GNC)_n_ gene formation was the formation of ss-(GNC)_n_ RNA with a random GNC codon sequence through a random process.

Thus, the synthesis of mature [GADV]-proteins composed of only [GADV]-amino acids became possible for the first time after the ds-(GNC)_n_ RNA gene was acquired and the translation system using four types of [GADV]-aa-tRNAs was established. Note that all members needed to execute both mRNA synthesis and the translational process had already been created at the point in time when the first ds-RNA gene was created, as both the use of the transcription and translation systems had already become possible, just after not the ds-(GNC)_n_ RNA gene, but ds-(GNC)_n_ RNA was formed [1] (Chapter 6).

##### The Emergence of Life

(1)Genuine life arose after the acquisition of various (GNC)_n_ genes through the creation of new homologous genes and entirely new (GNC)_n_ genes, which were derived from sense strands [36] and antisense strands [15] of previously established (GNC)_n_ genes, respectively [1] (Chapter 8).(2)The first life emerged not at one moment, but during consecutive changes. In other words, any critical moment of the emergence of the first life did not exist. The emergence of life would have been such a gradual change as it can be confirmed that a typical life had arisen after a point in time. Finally, ds-(GNC)_n_ RNA genes encoding a mature [GADV]-protein were formed as the most effective means for producing mature [GADV]-proteins with high catalytic activity. It should be noted again that the steps towards the formation of the first gene were the processes for selecting only [GADV]-amino acids more efficiently from chemically complex messy environments. Thus, the key to the first ds-(GNC)_n_ gene formation was the formation of ss-(GNC)_n_ RNA with a random GNC codon sequence through a random process.

## 4. Discussion

It is difficult to provide an answer to the question of how life emerged in “chemically complex messy environments”. The problem is essentially the same as the question of how the most primitive core life system, which is composed of [GADV]-protein, AntiC-SL tRNA (GNC primeval genetic code) and the (GNC)_n_ RNA gene using the respective small number of monomers, as four types of [GADV]-amino acids and four types of nucleotides, was established in those complex messy environments, because an answer to the mystery of the origin of life is, simultaneously, the answer to the question of how the small types of monomers were selected out from the chemically complex messy environments on primitive Earth. Therefore, it is necessary to consider focusing the discussion on how life arose using the core life system on primitive Earth in order to find the answer to the question of how life emerged in chemically complex messy environments.

The answer to the question is given based on the GADV hypothesis on the origin of life by considering several factors comprehensively, as described below.

Small types of, not so many, organic compounds including [GADV]-amino acids were able to be synthesized from inorganic compounds with prebiotic means on primitive Earth.[GADV]-polypeptides composed of mainly four types of [GADV]-amino acids were able to be produced through a random process on primitive Earth. This is because [GADV]-amino acids, having both positive and negative charges in the molecule, pulled against each other, owing to the electrostatic attraction. Immature [GADV]-proteins, actually aggregates of [GADV]-peptides, which were produced by the joining of [GADV]-amino acids randomly selected from [GADV]-amino acid composition or one of the protein 0th-order structures, were able to be folded into water-soluble globular structures with some flexibility to exhibit various weak but effective functions [26].

The reason the four types of [GADV]-amino acids were chosen as components of the most primitive protein is not only because the [GADV]-amino acids were able to be easily produced via prebiotic means and accumulated in large amounts on primitive Earth, but also because [GADV]-amino acids satisfy the four conditions (hydropathy, α-helix, β-sheet, turn/coil formabilities) for forming water-soluble globular proteins, which were obtained based on amino acid compositions [3,4]. In fact, it was confirmed that [GADV]-proteins, actually aggregates of [GADV]-peptides, which were obtained by repeated wet-dry cycles of [GADV]-amino acids, have various catalytic functions [26].

3.[GADV]-microspheres were able to be formed with immature [GADV]-proteins, actually [GADV]-peptide aggregates. The formation of [GADV]-microspheres, which demonstrated individuality, made it possible for growth, proliferation and evolution. It is supposed that lipids were incorporated into [GADV]-protein membranes to enhance the membrane function through an increase in membrane fluidity during the evolution of [GADV]-microspheres.4.Furthermore, metabolic pathways for [GADV]-amino acid synthesis using simple organic compounds as glyoxylate and pyruvate, which accumulated on primitive Earth, were able to be formed using immature [GADV]-proteins in the microspheres ([GADV]-protein world). The formation of the [GADV]-amino acid metabolic pathways assured the continuous growth, division and proliferation of [GADV]-microspheres through the synthesis of [GADV]-peptides.

It may be a matter of speculation whether or not [GADV]-amino acids were able to be synthesized with immature [GADV]-proteins using, as starting compounds, glyoxylate and pyruvate. The reason [GADV]-amino acid synthetic pathways were able to be formed is because such immature [GADV]-proteins, which have various catalytic activities or pluripotency as catalysts, could be used as biocatalysts [1] (Chapter 3) [26,27].

5.The metabolic pathway for the synthesis of four types of nucleotides was also able to be formed using immature [GADV]-proteins in [GADV]-microspheres, similar to the case of the synthetic pathways of [GADV]-amino acids.6.Further, it is considered that AntiC-SL tRNAs were formed with four types of nucleotides synthesized through metabolic pathways. Thus, the use of nucleotides were restricted for the first time into four, adenine (A), guanine (G), uracil (U) and cytosine (C), because two base pairs, A-U and G-C, were indispensable to folding RNA strands into AntiC-SL through hydrogen bonds with high directionality.7.Next, the GNC primeval genetic code, which determines the framework composed of four [GADV]-amino acids and four GNC codons, was established, although the corresponding relationships between [GADV]-amino acids and the GNC codons were accidentally determined and frozen, as assumed by the GNC code frozen-accident theory [1] (Chapter 7).8.Eventually, the first genuine life emerged on primitive Earth after (GNC)_n_ RNA genes were formed successively in order of ss-(GNC)_n_ RNA, ds-(GNC)_n_ RNA and the ds-(GNC)_n_ RNA gene, and the core life system was established.

Thus, the first genuine life using biopolymers composed of small types of components emerged in the chemically complex messy environments on primitive Earth approximately 4 billion years ago. I believe that such steps to the emergence of life were inevitable, and there was no way to the emergence of life. In this way, it can be reasonably explained, based on the GADV hypothesis on the origin of life, how the core life system is composed of the (GNC)_n_ gene, AntiC-SL tRNA and [GADV]-proteins, all of which are composed of small types of components, could be established in complex messy environments on primitive Earth. This indicates that the GADV hypothesis is a valid concept for explaining the steps to the emergence of life.

In addition, the GADV hypothesis is testable. Therefore, in order to further confirm the validity of the GADV hypothesis experimentally, I propose several experiments at the end of this article described below:A structural analysis and measurement of the various catalytic activities of a pluripotent immature [GADV]-protein with a random amino acid sequence;Confirmation of the growth, division and proliferation of [GADV]-microspheres, formed by the repeated wet-drying cycles of [GADV]-amino acids;Syntheses of [GADV]-amino acids and nucleotides with immature pluripotent [GADV]-proteins.Formation of AntiC-SL RNA during repeated random joining of nucleotides and its degradation.Formation of ss-(GNC)_n_ RNA by random joining of anticodons carried by four AntiC-SL RNAs.Establishment of the core life system accompanied by ds-(GNC)_n_ RNA gene, etc.

In the abstract of a recent review article about the Hot Spring Hypothesis on the origin of life of Damer and Deamer [37], it is described that “*a continuity is observed for biogenesis beginning with simple protocell aggregates, through the transitional form of the progenote, to robust microbial mats that leave the fossil imprints of stromatolites so representative in the rock record. A roadmap to future testing of the hypothesis is presented*”. I want to pay attention to the future development of the Hot Spring Hypothesis.

## Figures and Tables

**Figure 1 life-12-01319-f001:**
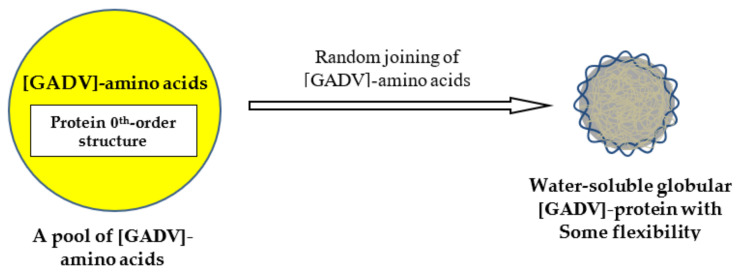
A polypeptide chain, which is obtained by joining amino acids randomly selected out from a pool (a protein 0th-order structure) containing roughly equal amounts of [GADV]-amino acids, should be folded into a water-soluble globular structure with some flexibility. The protein, actually aggregates of [GADV]-peptides, has a pluripotency that makes it possible to exhibit many catalytic activities owing to the structure flexibility. The wavy lines surrounding the gray circle and the thin yellow curved lines within the circle represent the flexible structure of an immature [GADV]-protein.

**Figure 2 life-12-01319-f002:**
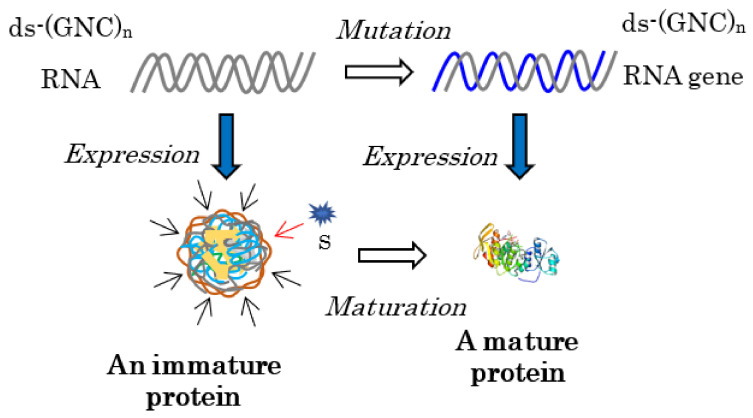
Every mature protein must be generated by the maturation of an immature protein, produced by the expression of an RNA strand encoding a random (GNC)_n_ sequence for the immature protein. This is because a substrate binding site (a key hole) must be formed as accumulating appropriate mutations to adjust the site to fit closely to a substrate (a key). For this purpose, the protein 0th-order structure is indispensable for producing an immature protein with some flexibility. The brown wavy lines surrounding the immature protein and the blue curved lines represent the flexible structure of the immature [GADV]-protein. The curved gray lines and the blue line indicate random (GNC)_n_ RNA strands and a (GNC)_n_ RNA gene encoding a mature [GADV]-protein, respectively.

**Figure 3 life-12-01319-f003:**
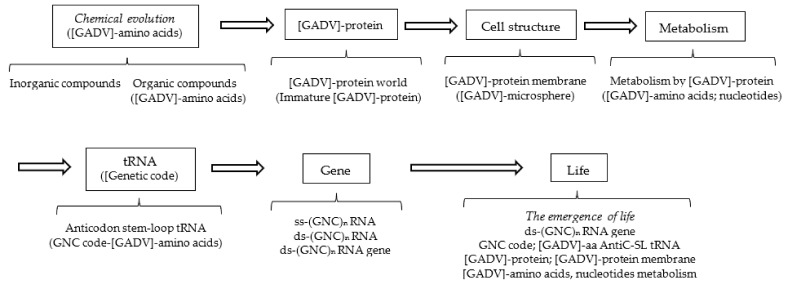
Possible steps from chemical evolution to the emergence of life, deduced from the [GADV]-protein world hypothesis (GADV hypothesis). All the steps are related to [GADV]-amino acids or [GADV]-protein, including GNC genetic code and the (GNC)_n_ gene, both of which encode [GADV]-amino acids and [GADV]-protein, respectively. The steps to the emergence of life can be reasonably explained using the GADV hypothesis, assuming that life emerged as the piling up of the six members (protein, cell structure, metabolism, tRNA, genetic code and gene) in order from [GADV]-protein to (GNC)_n_ gene encoding a mature [GADV]-protein.

**Figure 4 life-12-01319-f004:**
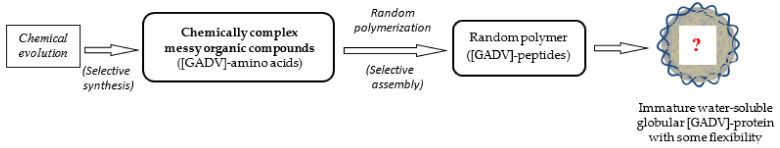
The organic compounds that were synthesized from inorganic compounds on primitive Earth via various prebiotic means were, naturally, chemically complex messy organic compounds. Therefore, a pool containing only [GADV]-amino acids did not exist on primitive Earth as expected by the GADV hypothesis. Therefore, one important question regarding how immature water-soluble globular [GADV]-proteins, which is a prerequisite in the GADV hypothesis, could be produced arises. Presenting an answer to the questions of how [GADV]-amino acids were selected and how immature [GADV]-protein could be formed is the main purpose of this article.

**Figure 5 life-12-01319-f005:**
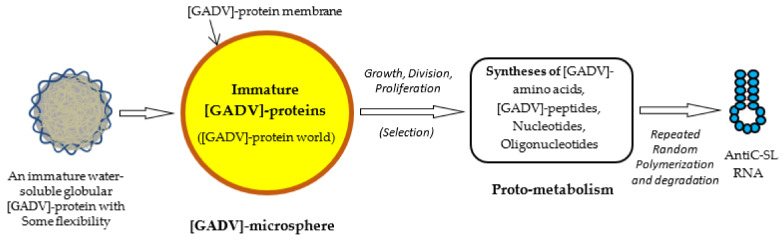
Possible steps from the formation of immature [GADV]-proteins to the formation of AntiC-SL RNA [1]. [GADV]-microspheres surrounded by [GADV]-protein membranes could be formed with [GADV]-peptides during repeated wet-drying processes. A [GADV]-protein world was formed in the [GADV]-microsphere. [GADV]-amino acids and nucleotides were synthesized by immature but pluripotent [GADV]-protein catalysts through proto-metabolism in the microsphere. Successively, AntiC-SL RNA was produced by the repeated random polymerization of nucleotides and their degradation. The core life system was established in [GADV]-microspheres, with a higher proliferation ability generated through the processes and, eventually, the first life arose on primitive Earth.

**Figure 6 life-12-01319-f006:**
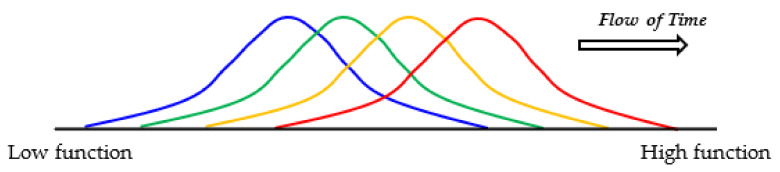
Evolution of [GADV]-microspheres without the genetic system. Random joining of amino acids carried out in the protein 0th-order structure inevitably generates [GADV]-peptides or [GADV]-proteins with a large distribution due to the random process. Therefore, at least part of [GADV]-proteins always had sufficiently high catalytic activities, which should have been effective for proceeding to the emergence of life. Thus, [GADV]-proteins with a higher catalytic activity than before could be generated in the microspheres. The steps to the emergence of life were the processes for the acquisition of [GADV]-proteins with higher catalytic activity than before, step-by-step. Changes in the distribution of [GADV]-microspheres from lower to higher functions are indicated in order from blue to red curves.

**Figure 7 life-12-01319-f007:**
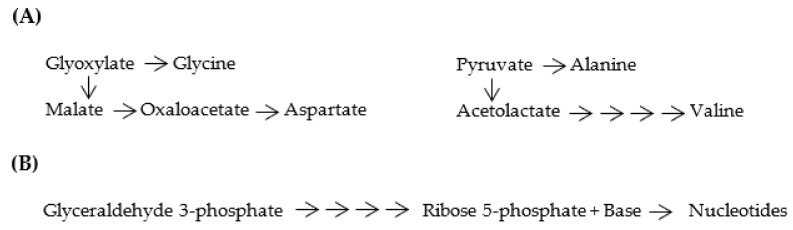
(**A**) Proto-metabolic pathways for [GADV]-amino acid synthesis. Four [GADV]-amino acids were produced from glyoxylate and pyruvate as starting materials through the proto-metabolic pathways. (**B**) Proto-metabolic pathways for nucleotide synthesis. Four nucleotides were produced with ribose 5-phosphate, which was synthesized from glyceraldehyde 3-phosphate as a starting material through the proto-metabolic pathways.

**Figure 8 life-12-01319-f008:**
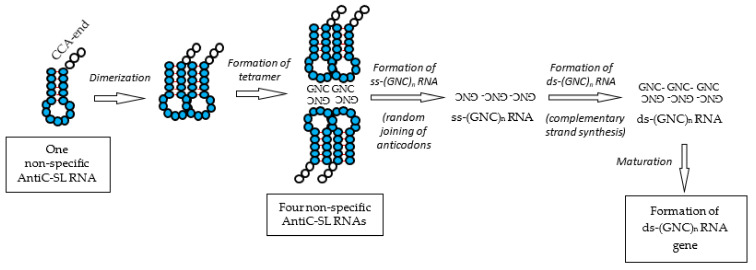
Formation process of the ds-RNA gene. The ds-RNA gene was formed through ss-(GNC)_n_ RNA and ds-(GNC)_n_ RNA. The ss-(GNC)_n_ RNA was formed by the random joining of GNC anticodons carried by AntiC-SL RNAs. The use of amino acids was restricted to four [GADV]-amino acids accompanied by the establishment of GNC primeval genetic code and the ds-(GNC)_n_ RNA gene for the first time.

**Table 1 life-12-01319-t001:** Steps from chemical evolution to the emergence of life, which are discussed in this article as dividing into three parts, I, II and III.

**Part I**: From chemical evolution to formation of [GADV]-microsphere
[GADV]-amino acid synthesis with prebiotic means Formation of [GADV]-microspheres ([GADV]-protein world) Selection and evolution of [GADV]-microsphere
**Part II**: After formation of [GADV]-microsphere to formation of AntiC-SL tRNAs
Formation of proto-metabolism in [GADV]-microsphere [GADV]-amino acid and [GADV]-peptide syntheses, Nucleotide synthesis with immature [GADV]-proteins (actually aggregates of [GADV]-peptides) Formation of AntiC-SL tRNAs
**Part III**: After formation of AntiC-SL tRNAs to the emergence of life
Formation of a ss-(GNC)_n_ RNA Formation of a ds-(GNC)_n_ RNA Formation of the first ds-(GNC)_n_ RNA gene The emergence of life

## Data Availability

Not applicable.

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
