# Peer review of "How Did Life Emerge in Chemically Complex Messy Environments?"

_life, 2022, doi:10.3390/life12091319_

Round 1

Reviewer 1 Report

1. In the abstract, the author addresses that there are three main points contributed to  open the way to the emergence of life. However, in the following statements, there are five points to be given, which is inconsistent.

2. What is the chemically messy environments? If does it contain phosphorus, minerals? It is better to give a general definition.

3. About line 179, why and how [GADV]-amino acid sequence was encoded by a random (GNC)n sequence on  one of two RNA strands? It need some experiments or references to support this statement. For example, Chem Rec 5: 107–118; 2005, this paper also need to cited here, though this paper has been cited in this manuscript.

4.About line 502, the statement "[GADV]-amino acids could be synthesized with simple organic compounds, as glyoxylate and pyruvate." need give out the supporting literature.

Besides, according to the Mill's experiment, fatty acid is hard to produce by the prebiotic means. Therefore, how can the usage of glyoxylate and pyruvate  be guaranteed on the primitive Earth?

5. In Fig 6, what is the meaning of each different colored curve? It is better to give some explanation.

6. The statement in line 673 need to give out the related literature.

7.  The [GADV]-protein world hypothesis has been proposed about 20 years since 2002 at the literature (Ikehara, K. Origins of gene, genetic code, protein and life: comprehensive view of life systems from a GNC-SNS primitive genetic code hypothesis. J Biosci 27, 165–186 (2002). https://doi.org/10.1007/BF02703773). If are there some experiments to be used to test the hypothesis?

Author Response

Reviewer 1

Thank you very much for your appropriate comments for my manuscript.

I reply to your valuable comments one by one, as described below.

  1. In the abstract, the author addresses that there are three main points contributed to  open the way to the emergence of life. However, in the following statements, there are five points to be given, which is inconsistent.

1) I have revised the phrase “three main points” to (Line 17): “The five factors described below” according to the indication of Reviewer 1.

  1. What is the chemically messy environments? If does it contain phosphorus, minerals? It is better to give a general definition.

2) I added the sentence in the abstract as described below.

Lines 15-17: In this article, the “chemically complex messy environments” that are focused on are a mixed state of various organic compounds produced via prebiotic means and accumulated on primitive earth.

  1. About line 179, why and how a random [GADV]-amino acid sequence was encoded by a random (GNC)n sequence on one of two RNA strands? It need some experiments or references to support this statement. For example, Chem Rec 5: 107–118; 2005, this paper also need to cited here, though this paper has been cited in this manuscript.

3) I added one sentence Lines 176-178: “An RNA with a random (GNC)n codon sequence could be formed by the random joining of GNC anticodons carried by AntiC-SL RNAs [1; Chapters 7 and 8].”

  1. About line 502, the statement "[GADV]-amino acids could be synthesized with simple organic compounds, as glyoxylate and pyruvate." need give out the supporting literature.

4) I cited one book chapter according to the Reviewer 1 as (Lines 512-515): because [GADV]-amino acids could be synthesized with simple organic compounds as glyoxylate and pyruvate----------[1; Chapter 5].

  1. Besides, according to the Mill's experiment, fatty acid is hard to produce by the prebiotic means. Therefore, how can the usage of glyoxylate and pyruvate be guaranteed on the primitive Earth?

5) I added the sentence as described below in the manuscript. Lines 352-356: “On the contrary, glyoxylate and pyruvate must be rather easily produced via prebiotic means, as those keto-acids carrying a small number of carbon atoms have active carbon atoms with a localized electron, although it must be confirmed that glyoxylate and pyruvate could be synthesized via prebiotic means.”

  1. In Fig 6, what is the meaning of each different colored curve? It is better to give some explanation.

6) I added the following sentence to explain the meaning of each different colored curve. Lines 492-493: “Changes in the distribution of [GADV]-microspheres from lower to higher functions are indicated in order from blue to red curves.”

  1. The statement in line 673 need to give out the related literature. 

7) I cited one book chapter Line 705: “ [1; Chapter 8]”, according to Reviewer 1 after (Figure 8).

  1. The [GADV]-protein world hypothesis has been proposed about 20 years since 2002 at the literature (Ikehara, K. Origins of gene, genetic code, protein and life: comprehensive view of life systems from a GNC-SNS primitive genetic code hypothesis. J Biosci27, 165–186 (2002). https://doi.org/10.1007/BF02703773). If are there some experiments to be used to test the hypothesis? 

8) I added the following paragraphs indicating my ideas at the last part of Discussion to answer to the above question of Reviewer 1.

Lines 851-864: In addition, the GADV hypothesis is testable. Therefore, in order to further confirm the validity of the GADV hypothesis experimentally, I propose the several experiments at the end of this article described below:

  1. A structural analysis and measurement of the various catalytic activities of a pluripotent immature [GADV]-protein with a random amino acid sequence;
  2. Confirmation of the growth, division and proliferation of [GADV]-microspheres, formed by the repeated wet-drying cycles of [GADV]-amino acids;
  3. Syntheses of [GADV]-amino acids and nucleotides with immature pluripotent [GADV]-proteins.
  4. Formation of AntiC-SL RNA during repeated random joining of nucleotides and its degradation.
  5. Formation of ss-(GNC)n RNA by random joining of anticodons carried by four AntiC-SL RNAs.
  6. Establishment of the core life system accompanied by ds-(GNC)n RNA gene, etc.

Reviewer 2 Report

Comments in the attached file.

Reviewer 3 Report

 The article under review is devoted to a consistent consideration of the GADV model of chemical evolution, culminating in the emergence of primary forms of life. This model that the author has been developing for many years is well founded and based on a large amount of objective data in the field of organic chemistry. The important conclusion is well argued that the primary molecules of the RNA world by themselves cannot serve as the initial basis for the emergence of the simplest forms of life, and their evolution is possible only through interaction first with immature, and then with mature proteins. At the same time, in my opinion, the role of lipids in the prebiological process described by the author seems to be too insignificant, and can be discussed within the framework of scientific discussion. But this does not reduce the value of this work, which is recommended for publication. From technical notes: deciphering the abbreviation GADV is desirable not only in the article (where it is), but also in the abstract, since not all scientists involved in interdisciplinary studies on the origin of life are professionals in organic chemistry.

Author Response

Reviewer 3

The article under review is devoted to a consistent consideration of the GADV model of chemical evolution, culminating in the emergence of primary forms of life. This model that the author has been developing for many years is well founded and based on a large amount of objective data in the field of organic chemistry. The important conclusion is well argued that the primary molecules of the RNA world by themselves cannot serve as the initial basis for the emergence of the simplest forms of life, and their evolution is possible only through interaction first with immature, and then with mature proteins. At the same time, in my opinion, the role of lipids in the prebiological process described by the author seems to be too insignificant, and can be discussed within the framework of scientific discussion. But this does not reduce the value of this work, which is recommended for publication. From technical notes: deciphering the abbreviation GADV is desirable not only in the article (where it is), but also in the abstract, since not all scientists involved in interdisciplinary studies on the origin of life are professionals in organic chemistry.

1) I added the sentence to explain my idea about the “role of lipid” as below.

Lines 810-813: It is supposed that lipids were incorporated into [GADV]-protein membranes to enhance the membrane function through an increase in membrane fluidity during the evolution of [GADV]-microspheres.

2) Furthermore, according to the indication of Reviewer 3, I added the sentence below in the abstract to explain the abbreviation [GADV].

Lines 19-21: [GADV] stands for four amino acids, Gly [G], Ala [A], Asp [D] and Val [V], which are indicated by a one-letter symbol.

Author Response

Reviewer 4

The review article "How did Life emerge in Chemically Complex Messy Environments?" illustrates the author theory of the "[GADV]-protein " world for elucidate the origin of life mystery. The author hypothesis could be interesting but I think that it is weak in too many aspects:

  • no one experiment is shown to support the [GADV]-protein world model, so my feeling is to deal more with a philosophical essay rather than a scientific review

1) The studies on the [GADV]-protein world model were carried out mainly with analyses of databases of genes, proteins, tRNAs and so on, which were obtained by experiments. Furthermore, I would like to advocate that it is quite important to conduct theoretical studies in advance, especially in the case of the origin-life studies, although, of course, experiments are quite important even in the research field. Therefore, I believe that my considerations described in the manuscript is a scientific review rather than a philosophical essay. Of course, I realize the importance of experiments. So, I present several experiments in the last part of “Discussion” of this manuscript as described below.

Lines 851-864

In addition, the GADV hypothesis is testable. Therefore, in order to further confirm the validity of the GADV hypothesis experimentally, I propose the several experiments at the end of this article described below:

  1. A structural analysis and measurement of the various catalytic activities of a pluripotent immature [GADV]-protein with a random amino acid sequence;
  2. Confirmation of the growth, division and proliferation of [GADV]-microspheres, formed by the repeated wet-drying cycles of [GADV]-amino acids;
  3. Syntheses of [GADV]-amino acids and nucleotides with immature pluripotent [GADV]-proteins.
  4. Formation of AntiC-SL RNA during repeated random joining of nucleotides and its degradation.
  5. Formation of ss-(GNC)n RNA by random joining of anticodons carried by four AntiC-SL RNAs.
  6. Establishment of the core life system accompanied by ds-(GNC)n RNA gene, etc.

In addition, my idea about points at issue of experiment-centered research in the field of the origin of life is described in (Lines 74-120) Section “B. The problem of a research strategy based on experiments” in Introduction. 

  • the choice of the four GADV amino acids that support all the [GADV]-protein hypothesis is not satisfactorily demonstrated.

2) I added one paragraph in the manuscript (Lines 797-804) to explain the reason why I chose the four GADV amino acids.

Lines 801-808

The reason the four types of [GADV]-amino acids were chosen as components of the most primitive protein is not only because the [GADV]-amino acids were able to be easily produced via prebiotic means and accumulated in large amounts on primitive Earth, but also because [GADV]-amino acids satisfy the four conditions (hydropathy, a-helix, b-sheet, turn/coil formabilities) for forming water-soluble globular proteins, which were obtained based on amino acid compositions [3, 4]. In fact, it was confirmed that [GADV]-proteins, actually aggregates of [GADV]-peptides, which were obtained by repeated wet-dry cycles of [GADV]-amino acids, have various catalytic functions [26].  

  • the author claims (lines 61-62) that gene and protein with an ordered sequence never be formed through a random process. This concept is reaffirmed throughout the paper by using the word messy. I think that the author underestimates the role of the Darwinian molecular selection during the evolution from the primordial chemical precursors to the modern cell

3) I consider as stated by Reviewer 4 that (Lines 66-68) it is easily supposed that “a mature gene and a mature protein with an ordered sequence could never be formed through a random process at one stroke,” Along with that, I also described in the manuscript (Lines 179-183) that “Specifically, every gene that encodes a mature protein has been formed as a result of maturation from an immature or incomplete water-soluble globular protein with some flexibility, which generates various weak catalytic activities or demonstrates pluripotency [1; Chapter 3], to a mature protein with a rigid structure and high catalytic activity.” Therefore, my idea about the formation of a mature gene and a mature protein is similar to the formation under the Darwinian molecular evolution, which is indicated by Reviewer 4.

  • the experimental knowledge of the actual prebiotic chemistry and molecular evolution is not adequately presented (for instance the formamide model is completely ignored even if more than 4000 papers are present in the international literature)

4) I added two paragraphs to explain the prebio chemistry for synthesis of nucleotides as shown below.

Lines 567-572; 573-578

It is still controversial whether or not nucleotides and nucleosides were actually synthesized using the reactants of simple inorganic compounds via prebiotic means, although some experimental results showing that nucleotides and nucleosides could be produced with ribose-5-phspate or ribose and nucleobases, respectively, have been reported [9, 29, 30], because the concentrations and purity of the reactants used in the experiments are usually quite different from the conditions on primitive Earth.

Even papers showing that nucleosides and nucleotides were synthesized from formamide with meteorite catalysts under proton irradiation, were published [31, 32]. Nevertheless, nucleotides must be depleted from surroundings of [GADV]-microspheres, which proliferated exponentially. This means that proto-metabolism for nucleotide synthesis must be established in [GADV]-microspheres before deprivation of nucleotides.

  • I believe that the aspect of novelty of this paper is completely missing and that it not add anything more to the book cited by the author himself (lines 118-121)

5) I added one paragraph to explain the aspect of novelty of this manuscript as describe below.

Lines 133-138    

Therefore, this article is described as providing an answer to the question that has been proposed in the Special issue: "Origin of Life in Chemically Complex Messy Environments". In other words, I discuss in the article how a small types of amino acids or nucleotides were selected from the chemically complex messy environments of primitive Earth based on the GADV hypothesis, which I propose. The answer to the question described in the article is the novelty aspect of this paper.

  • some author statements are contradicted by published experimental results (for instance lines 352-356 are contradicted by the demonstration that syntheses of nucleosides are meteorite-catalyzed from formamide under proton irradiation (Saladino et al. https://doi.org/10.1073/pnas.142222511) and that nucleotides can form successively in the presence of phosphate minerals (Costanzo et al. JBC, 282,2007).

6) I cited the papers, Saladino et al. and Costanzo et al. shown by Reviewer 4 in the revised manuscript (Lines 573-578).

Even papers showing that nucleosides and nucleotides were synthesized from formamide with meteorite catalysts under proton irradiation, were published [31, 32]. Nevertheless, nucleotides must be depleted from surroundings of [GADV]-microspheres, which proliferated exponentially. This means that proto-metabolism for nucleotide synthesis must be established in [GADV]-microspheres before deprivation of nucleotides.

7) the geochemical contest in which the life could arise is completely ignored.

7) I do not ignore the geochemical context, because messy environments composed of various organic compounds were formed via prebiotic means containing geochemical reactions on primitive Earth. That is described in Section 1. from Line 313: Preferential synthesis of [GADV]-amino acids via prebiotic means.   

Round 2

Reviewer 4 Report

I have read the revised maniscript and I agree for the publication.